# Tear Film Osmolarity Measurement in Japanese Dry Eye Patients Using a Handheld Osmolarity System

**DOI:** 10.3390/diagnostics10100789

**Published:** 2020-10-05

**Authors:** Jun Shimazaki, Miki Sakata, Seika Den, Miki Iwasaki, Ikuko Toda

**Affiliations:** 1Department of Ophthalmology, Tokyo Dental College, Ichikawa General Hospital, Chiba 272-8513, Japan; 2Shinjuku Eye Clinic, Tokyo 169-0074, Japan; sakata1788@gmail.com; 3Shimazaki Eye Clinic, Tokyo 100-0014, Japan; denseika@gmail.com; 4Department of Ophthalmology, The Jikei University School of Medicine, Tokyo 105-8461, Japan; 5Ryogoku Eye Clinic, Tokyo 130-0026, Japan; mikiwasaki@aol.com; 6Minamiaoyama Eye Clinic, Tokyo 107-0061, Japan; toda@minamiaoyama.or.jp

**Keywords:** dry eye disease, tear film osmolarity, tear film break-up pattern

## Abstract

We studied the efficacy and safety of a handheld osmolarity measurement system (I-PEN) in Japanese patients with dry eye disease (DED) and non-DED subjects. In this prospective, multicenter study, tear osmolarity was examined using the I-PEN in a total of 122 eyes divided into DED (*n* = 71) and non-DED (*n* = 51) groups. Subjective symptoms were assessed using the Dry Eye-Related Quality-of-Life Score (DEQS) questionnaire. Ocular surface condition was evaluated in terms of fluorescein tear breakup time (FBUT) and tear breakup pattern (TBUP), and by fluorescein staining and Schirmer’s test. The I-PEN measurements were performed safely in the majority of cases. There was no statistically significant difference in mean tear film osmolarity between the DED and non-DED groups (294.76 ± 16.39 vs. 297.76 ± 16.72 mOsms/L, respectively, *p* = 0.32). No significant correlations were observed between osmolarity values and DEQS score, FBUT, or the Schirmer score. Osmolarity did not differ among TBUP subgroups. This prospective clinical study found no correlations between the tear film osmolarity values obtained with the I-PEN system and any subjective or objective parameters of DED. Further studies are required to determine the utility of the I-PEN system in other settings.

## 1. Introduction

Dry eye disease (DED) is one of the most common ophthalmic disorders in Japan, with a reported prevalence of 12.5% in men and 21.6% in women aged over 40 years [1,2]. Proper diagnosis and examination are essential for physicians to provide adequate treatment to patients. The main diagnostic tests are tear secretion tests, such as Schirmer’s test, the vital staining test, and the tear stability test, which includes a measurement of fluorescein tear breakup time (FBUT). However, these tests do not have sufficient sensitivity and specificity, and more reliable biomarkers are therefore needed. Tear osmolarity is one of the most promising biomarkers for diagnosis and monitoring of DED [3,4,5], and is considered to be involved in the pathogenesis of DED [6]. Hyperosmolarity of tears is considered to cause inflammation and damage of the ocular surface, irritating symptoms, and compensatory mechanism in DED. Several reports demonstrated that the hyperosmolarity stimulates various inflammatory events in the ocular surface epithelia such as generation of inflammatory cytokines and matrix metalloproteinases (MMPs) [7,8]. While a number of clinical studies based on osmolarity have been conducted, the diagnostic value of this measurement remains controversial [9,10].

The I-PEN^®^ Osmolarity System (I-PEN; I-MED Pharma Inc., Dollard-des-Ormeaux, QC, Canada) is a handheld, solid-state electronic diagnostic device for the quantitative measurement of tear osmolarity. The I-PEN is used in conjunction with single use sensors (SUS) and constitutes a rapid (~2 s) and simple method for determining tear osmolarity based on analysis of palpebral conjunctiva tissues bathed in tear films. Here, we prospectively studied the efficacy and safety of the I-PEN for use in healthy controls and DED patients in a clinical setting. The study had two specific purposes: to assess whether the I-PEN is useful for diagnosing Japanese DED patients, and to determine its usefulness in subgroups of DED patients classified according to tear film breakup pattern (TBUP).

## 2. Materials and Methods

This prospective, non-randomized, multicenter study involved five ophthalmic clinics in the Tokyo metropolitan area of Japan. This research was performed in accordance with the tenets of the Declaration of Helsinki, and the study was approved by the internal review board of the Ryogoku Eye Clinic (Ryogoku 2017, 19 December 2017). We planned to examine 50 eyes in 50 consecutive subjects in each clinic. The exclusion criteria were age <20 years, current contact lens wearer, any ocular surface disease other than DED, current eye drop use, use of systemic medications that can influence tear/ocular surface condition, pregnancy, and not consenting to participate in the study.

Routine ophthalmic measurements (including visual acuity and non-contact tonometry), subjective symptoms, fluorescein staining, FBUT, and TBUP were assessed, in that order (to prevent interference effects). Schirmer’s test was performed at least 5 min after the I-PEN measurements. After inserting in the I-PEN, the tip of the SUS was gently placed on the inferior palpebral conjunctiva trying not to immerse in the tear meniscus. No local anesthetics were used. While both eyes of the subjects were examined, only the results of the eyes with the shorter FBUT were included in the analysis. If the eyes had the same FBUT value, the data of the right eye were used. 

Differences in I-PEN values between the non-DED and DED groups were examined. The relationships between the I-PEN values and subjective symptoms, FBUT, TBUP, fluorescein staining score, and the Schirmer score were studied. I-PEN measurement failures (e.g., due to SUS breakage) were recorded. Subjective patient discomfort during measurements was also recorded.

The I-PEN evaluations were performed by certified ophthalmic examiners after receiving instruction from skilled operators. First, the SUS was inserted into the I-PEN. Then, patients were asked to gently close their eyes for approximately 30 s. On opening their eyes, the tip of the SUS was placed directly onto the inferior palpebral conjunctiva at an angle of 45°, with both gold nodes of the SUS in good contact with the conjunctiva. After several seconds, the I-PEN emitted an audible beep and displayed the osmolarity value. Discomfort during the measurements was rated on a scale ranging from 1 to 5 (1, no discomfort; 5, severe pain).

Subjective symptoms of DED were assessed using the Dry Eye-Related Quality-of-Life Score (DEQS) questionnaire, which consists of 15 questions and yields a single summary score [11]. The summary score provides a quantitative measure of DED symptoms; scores of 0 and 100 indicate the best and worst, respectively. Test strips containing fluorescein sodium (Fluores ocular examination test paper; Ayumi Pharmaceutical Co., Tokyo, Japan) were used for fluorescein staining and FBUT measurement. Fluorescein staining scores were obtained for the temporal bulbar conjunctiva, nasal bulbar conjunctiva, and cornea (0–3 points; 0, no damage; 3, damage over the entire area) [12]. FBUT was measured three times consecutively, and the average value was calculated. TBUT ≤ 5 s was regarded as reduced tear film stability. 

The TBUP test is used to assess dynamic patterns of tear film breakup [13]. TBUP can be classified into five different breakage patterns: area, line, spot, dimple, or random break. Each pattern is considered to reflect the underlying DED pathophysiology. DED was classified into three categories based on the underlying pathophysiology responsible for tear film instability: aqueous-deficient (area or line break), decreased surface wettability (spot or dimple breaks), or excessive evaporation (random break) [14]. 

The Asia Dry Eye Society diagnostic criteria for DED were used. Briefly, symptomatic eyes with FBUT ≤ 5 s were regarded as having DED [15]. A DEQS cutoff value of 15 was used to classify subjects as positive or negative for symptoms, as reported previously [16].

All statistical analyses were performed with R software (ver. 3.6.3; R Project for Statistical Computing, Vienna, Austria). A two-sample *t*-test was used to compare osmolarity between the non-DED and DED subjects, and the Tukey–Kramer test was used to compare osmolarity between non-DED and DED eyes. Values are presented as mean ± SD unless otherwise indicated. In all analyses, *p* < 0.05 was taken to indicate statistical significance.

## 3. Results

### 3.1. Profile of Study Subjects

We recruited a total of 144 subjects between April 2018 and March 2019. We excluded 22 subjects because of insufficient data; thus, the analyses were performed on 122 eyes. There were 71 DED eyes and 51 non-DED eyes. The characteristics of the study subjects are presented in Table 1. The aqueous-deficient pattern (area or line breaks) was the most common TBUP (58 eyes; 36 and 22 DED and non-DED eyes, respectively), followed by excessive evaporation (random breaks, 48 eyes; 25 and 23 DED and non-DED eyes, respectively) and decreased wettability (spot or dimple breaks, 13 eyes; 8 and 5 DED and non-DED eyes, respectively). Two eyes could not be classified according to TBUP.

### 3.2. Tear Film Osmolarity and DED

Tear film osmolarity showed a normal distribution, with a mean value of 296.02 ± 16.52 mOsms/L. There were no significant interexaminer differences in the Generalized Linear Mixed Model. There was no significant difference in mean tear film osmolarity between the DED and non-DED subjects (294.76 ± 16.39 vs. 297.76 ± 16.72 mOsms/L, respectively, *p* = 0.32; Figure 1).

### 3.3. Tear Film Osmolarity and Other Parameters

#### 3.3.1. Osmolarity and Subjective Symptoms

The mean DEQS summary score was 28.52 ± 21.49 and was significantly higher in the DED group than the non-DED group (40.45 ± 18.43 vs. 11.92 ± 12.63, respectively, *p* < 0.0001). There was a trend toward a negative correlation between the DEQS summary score and tear osmolarity value (*r* = −0.168, *p* = 0.064, Pearson’s correlation test; Figure 2).

#### 3.3.2. Osmolarity and FBUT or TBUP

There was no difference in osmolarity between eyes with an FBUT > 5 s (*n* = 14) and ≤ 5 s (*n* = 108; 299.86 ± 15.48 vs. 295.67 ± 16. 93 mOsms/L, respectively, *p* = 0.38). Osmolarity tended to decrease with shorter FBUT (*r* = 0.142, *p* = 0.118, Pearson’s correlation test; Figure 3).

The osmolarity values in eyes with the aqueous-deficient, decreased wettability, and excessive evaporation TBUP were 295.89 ± 18.12, 289.38 ± 21.33, and 295.76 ± 12.22 mOsms/L, respectively (*p* > 0.05, Tukey–Kramer test; Figure 4).

#### 3.3.3. Osmolarity and Schirmer Test Score

There was no significant difference in tear film osmolarity between eyes with a Schirmer test score >5 mm and ≤5 mm (295.55 ± 16.00 vs. 289.89 ± 19.88 mOsms/L, respectively, *p* = 0.13).

#### 3.3.4. Osmolarity and Fluorescein Score

The mean fluorescein score in all eyes examined was 0.09 ± 1.39 points, and there was no correlation between tear film osmolarity and fluorescein staining score (*r* = 0.011, *p* = 0.902, Pearson’s correlation test; Table 2).

### 3.4. I-PEN Measurement Failures and Patient Discomfort

Among the 144 subjects examined, measurement failures occurred in 18 cases. Most failures were attributed to instrument breakage, which was addressed by replacing the SUS of the I-PEN. The mean discomfort level was 1.54 (0.68); 92% of the subjects felt no or slight discomfort and only 2% reported severe pain (Table 3).

## 4. Discussion

There have been remarkable advances in basic and clinical DED research over the last few decades. Despite these advances, however, a lack of reliable and easily applicable DED biomarkers remains problematic. A number of reports of tear cytokine/chemokine measurements and proteomics have appeared [17,18,19]. Tear osmolarity has been regarded as the most reliable DED biomarker, especially when measured with the TearLab^®^ Osmolarity System (TearLab, Escondido, CA, USA). A number of clinical studies have demonstrated the efficacy of osmolarity for diagnosing, grading, and managing the treatment of DED [3,4,5]. However, recent reports demonstrated poor reliability and repeatability of TearLab^®^ osmolarity measurements, and the usefulness of the system remains controversial [9,10,20].

For the TearLab system, a tear osmolarity value of 302 mOsm/L is considered normal, with minimal inter-eye difference. Thus, 308 mOsm/L (in either eye) is often used as the threshold for differentiating between normal and early stage DED cases, with 316 mOsm/L used as a cutoff for more advanced DED [21]. The mean osmolarity value in the present study was somewhat lower than in previous studies using the TearLab system. The reason for this discrepancy is not clear, but it may be attributable to a difference in measurement site (inferior conjunctival cul-de-sac for I-PEN vs. inferior tear meniscus for TearLab). McMonnies proposed a model in which osmolarity is lowest in the upper conjunctival sac and increases progressively in the upper meniscus, upper part of the exposed ocular surface, lower part of the exposed ocular surface, lower meniscus, and lower conjunctival sac [22].

In the present study, tear osmolarity did not differ between eyes with and without DED. In addition, we found no correlations between osmolarity and subjective symptom scores or objective DED parameters. In fact, osmolarity tended to decrease as symptom scores and TBUT worsened, although these associations were not statistically significant (Figure 2 and Figure 3). We also hypothesized that tear osmolarity may be valuable in certain subtypes of DED. However, we did not find any differences in osmolarity values among the TBUP subgroups (Figure 4). 

It may be too early to draw a conclusion that the measurement of tear osmolarity has no clinical values as there are several possible explanations for the negative findings in the present study. First, it should be noted that the majority of the eyes in the DED group were classified as mild severity based on the FBUT and Schirmer score (Table 1). A study including moderate to severe DED cases would be better able to determine the clinical feasibility of I-PEN measurements. Second, osmolarity measurements obtained using the I-PEN system may not be accurate, although Chan et al. reported that the I-PEN system yielded rapid and accurate measurements of tear osmolarity in their “simulated testing setting” [23]. In contrast, some studies comparing different osmolarity measurement systems reported inferior repeatability of I-PEN measurements compared to those of other instruments [24,25]. It is possible that contact between the SUS of the I-PEN and the tarsal conjunctiva, even if extremely slight, may induce transient lacrimation, resulting in a decrease in tear osmolarity. The present study involved multiple examiners as it was a multicenter study. The staff attended lectures pertaining to the correct method for application of the instrument, and there was no significant variability in measurements among the clinics. However, operator error resulting in irritation and lacrimation cannot be ruled out. Other possible sources of lacrimation included routine ophthalmic examinations performed prior to the I-PEN measurements. In particular, light from the slit lamp may affect osmolarity values, although additional prospective studies using different study protocols are needed to verify this.

## 5. Conclusions

In summary, this prospective clinical study found no correlations between tear film osmolarity values obtained with the I-PEN system and any subjective or objective parameters of DED. We did not find any differences in tear film osmolarity in different subtypes of DED determined by tear breakup patterns. Further studies are required to determine the utility of the I-PEN system in other settings. 

## Figures and Tables

**Figure 1 diagnostics-10-00789-f001:**
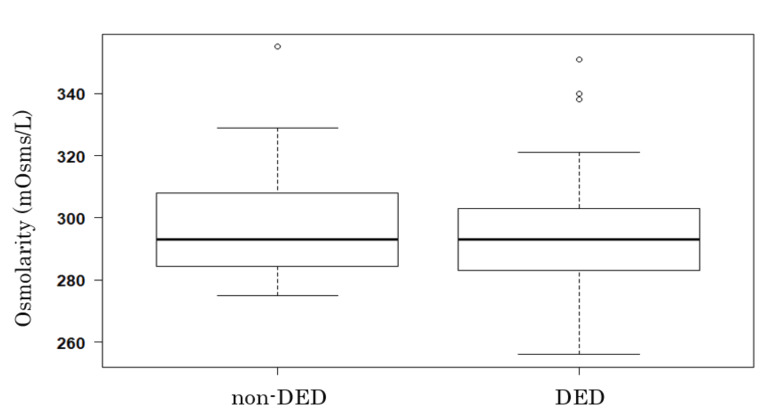
The mean osmolarity values of non-dry eye disease (non-DED; *n* = 51) and DED (*n* = 71) eyes were 297.76 ± 16.72 and 294.76 ± 16.39 mOsms/L, respectively (*t*-test, *p* = 0.32). Bold line denotes median, and upper and lower ends of the box denote 75th and 25th percentile, respectively. Upper and lower error bars denote maximum and minimum values, respectively.

**Figure 2 diagnostics-10-00789-f002:**
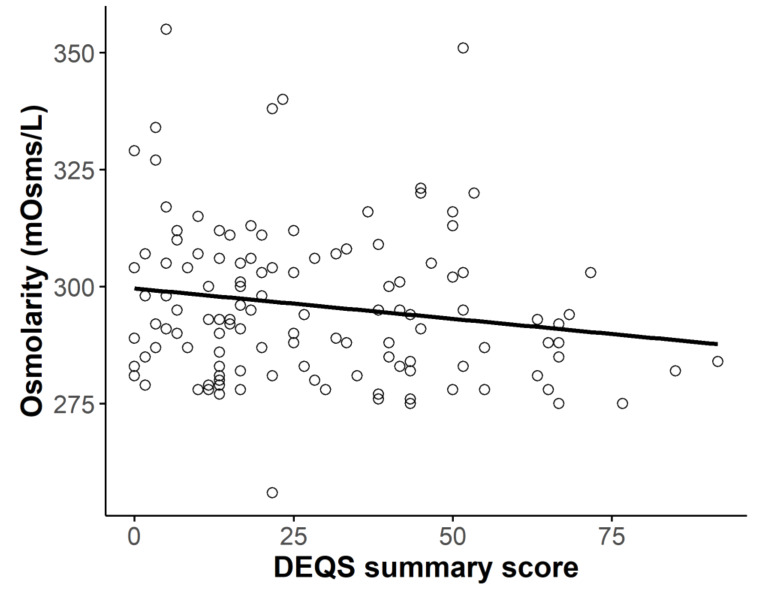
Relationship between total Dry Eye-Related Quality-of-Life Score (DEQS) and osmolarity. There was no correlation between total DEQS score and osmolarity (*r* = −0.168, *p* = 0.064, Pearson’s correlation test).

**Figure 3 diagnostics-10-00789-f003:**
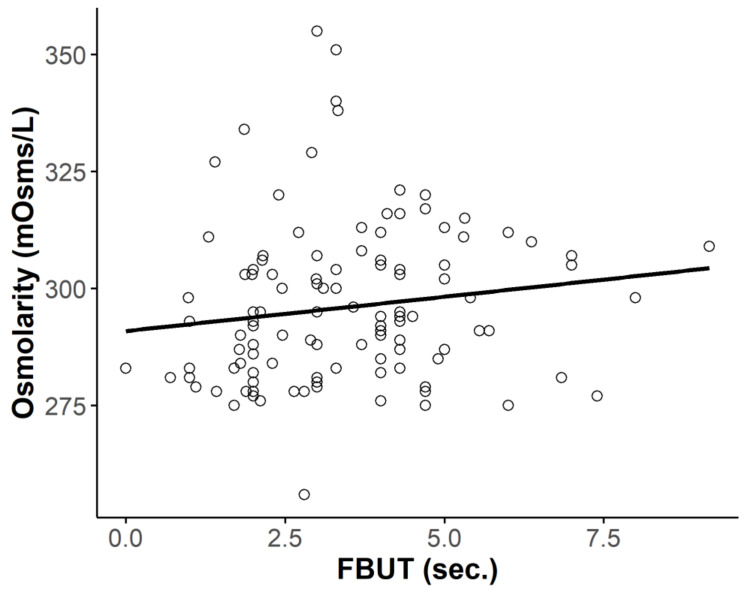
Relationship between total fluorescein tear breakup time (FBUT) and osmolarity. There was no correlation between FBUT and osmolarity (*r* = 0.142, *p* = 0.118, Pearson’s correlation test).

**Figure 4 diagnostics-10-00789-f004:**
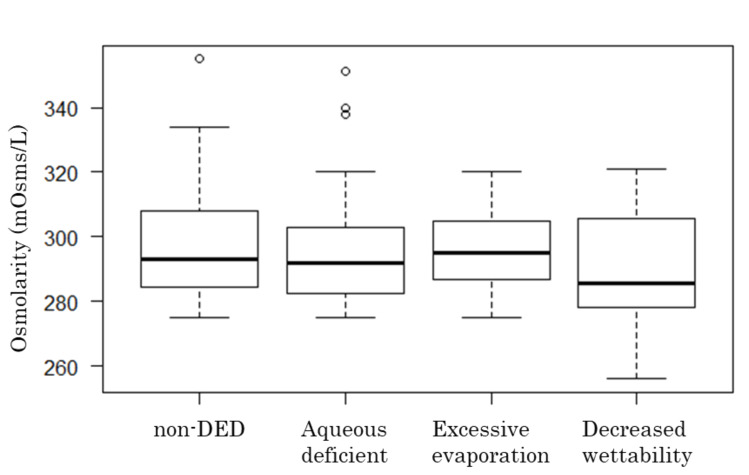
Comparison of osmolarity values among tear breakup pattern (TBUP) subgroups. There was no significant difference in osmolarity among the groups (*p* > 0.05, Tukey–Kramer test). Bold line denotes median, and upper and lower ends of the box denote 75th and 25th percentile, respectively. Upper and lower error bars denote maximum and minimum values, respectively.

**Table 1 diagnostics-10-00789-t001:** Characteristics of the study subjects.

Profile of the Subjects	DED ^1^ (Mean ± SD ^2^)	Non-DED (Mean ± SD)	Total (Mean ± SD)
Sex (male:female)	18:53	10:41	28:94
Age (years)	45.39 ± 13.88	44.82 ± 18.73	44.74 ± 16.03
DEQS ^3^ summary score	40.45 ± 18.43	11.92 ± 12.63	28.52 ± 21.49
FBUT ^4^ (s)	3.12 ± 1.15	3.82 ± 2.04	3.41 ± 1.61
FBUT by TBUP ^5^ subgroup(s)			
Aqueous-deficient	2.64 ± 0.89 (*n* = 36)	2.97 ± 2.1 (*n* = 22)	2.77 ± 1.48 (*n* = 58)
Decreased wettability	2.59 ± 1.17 (*n* = 8)	3.37 ± 2.06 (*n* = 5)	2.89 ± 1.54 (*n* = 13)
Excessive evaporation	4.08 ± 0.85 (*n* = 25)	4.83 ± 1.50 (*n* = 23)	4.44 ± 1.25 (*n* = 48)
Osmolarity by TBUP subgroup (mOsms/L)	294.76 ± 16.39	297.76 ± 16.72	296.02 ± 16.52
Aqueous-deficient	295.89 ± 18.12 (*n* = 36)	299.73 ± 16.17 (*n* = 22)	297.35 ± 17.36 (*n* = 58)
Decreased wettability	289.38 ± 21.33 (*n* = 8)	292.60 ± 17.27 (*n* = 5)	290.62 ± 19.17 (*n* = 13)
Excessive evaporation	295.76 ± 12.22 (*n* = 25)	297.35 ± 17.86 (*n* = 23)	296.52 ± 15.04 (*n* = 48)
Schirmer test score (mm/5 min)	15.04 ± 10.05	19.51 ± 12.24	16.91 ± 11.19

^1^ DED, dry eye disease; ^2^ SD, standard deviation ^3^ DEQS, Dry Eye-Related Quality-of-Life Score; ^4^ FBUT, fluorescein breakup time; ^5^ TBUP, tear film breakup pattern.

**Table 2 diagnostics-10-00789-t002:** Associations between tear osmolarity and other parameters.

Parameters	Correlation Coefficient	*p*-Value
Fluorescein score	0.011	0.902
FBUT ^1^	0.142	0.118
Schirmer’s value	0.036	0.696
DEQS ^2^ Summary score	−0.168	0.064

^1^ FBUT, fluorescein break-up time; ^2^ DEQS, Dry Eye-Related Quality-of-Life Score.

**Table 3 diagnostics-10-00789-t003:** Measurement failures and discomfort level of I-Pen measurement.

Adverse Event	Numbers of Adverse Event and Discomfort Level
Number of measurements failures	18
Succeeded on second attempt	5
Succeeded after multiple attempts	6
Failed (no data)	7
Discomfort level	
1: No pain	78
2: Slight pain	55
3: Moderate pain	7
4: Severe pain	3
5: Extremely severe pain	0
No response	1

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
