# Peer review of "Tear Film Osmolarity Measurement in Japanese Dry Eye Patients Using a Handheld Osmolarity System"

_diagnostics, 2020, doi:10.3390/diagnostics10100789_

Round 1
Reviewer 1 Report
the authors are to be commended for an thorough review. unfortunately, there seems to be no clinical significance to the device or the work. back to the drawing board
Author Response
We thank the reviewer for the kind comment. We added the sentence in Discussion suggesting further studies are needed to demonstrate the usefulness of the tear osmolarity measurement using I-PEN system.
line 221 "It may be too early to draw conclusion that the measurement of tear osmolarity has no clinical values as there are several possible explanations for the negative findings in the present study. "

Reviewer 2 Report
the manuscript is original and well structured, the introductory part must be expanded by specifying the inflammatory state that is present in the dry eye as it modifies the osmolarity.
I also recommend that you insert some bibliographic entries in this chapter.
Szczesna-Iskander DH. Measurement variability of the TearLab Osmolarity System. Cont Lens Anterior Eye. 2016;39(5):353-358. doi:10.1016/j.clae.2016.06.006
Scorolli L, Meduri A, Morara M, et al. Effect of cysteine in transgenic mice on healing of corneal epithelium after excimer laser photoablation. Ophthalmologica. 2008;222(6):380-385. doi:10.1159/000151691
Baiocchi S, Mazzotta C, Sgheri A, et al. In vivo confocal microscopy: qualitative investigation of the conjunctival and corneal surface in open angle glaucomatous patients undergoing the XEN-Gel implant, trabeculectomy or medical therapy. Eye Vis (Lond). 2020;7:15. Published 2020 Mar 10. doi:10.1186/s40662-020-00181-8
also expand the materials and methods and improve the English language (with a native speaker technician). in the improved conclusions the scientific goal to be obtained
Author Response
We thank the reviewer for the suggestion.
- We expanded the explanation on tear hyperosmolarity, especially its relationship with ocular surface inflammation.
line 47-51 "Hyperosmolarity of tears is considered to cause inflammation and damage of the ocular surface, irritating symptoms, and compensatory mechanism in DED. Several reports demonstrated that the hypeosmolarity stimulates various inflammatory events in the ocular surface epithelia such as generation of inflammatory cytokines and MMPs[7, 8]. "
2. We added the paper by Szczesna-Iskander in Discussion (line 202).
We also conducted an English editing by native speakers.

Round 2
Reviewer 1 Report
Unfortunately, this work does not add to the body of knowledge in a significant way.
Reviewer 2 Report
the changes made have improved the quality of the manuscript, making it more complete, flowing.
in my opinion I think it can be published